# Far-Red Light Effects on Lettuce Growth and Morphology in Indoor Production Are Cultivar Specific

**DOI:** 10.3390/plants11202714

**Published:** 2022-10-14

**Authors:** Jun Liu, Marc W. van Iersel

**Affiliations:** 1Crop Physiology Laboratory, Utah State University, Logan, UT 84322, USA; 2Horticultural Physiology Laboratory, Department of Horticulture, University of Georgia, Athens, GA 30602, USA

**Keywords:** far-red photons, leaf elongation, light interception, photosynthesis, shade acclimation, vegetable production, *Lactuca sativa* L.

## Abstract

Understanding crop responses to the light spectrum is critical for optimal indoor production. Far-red light is of particular interest, because it can accelerate growth through both physiological and morphological mechanisms. However, the optimal amount of supplemental far-red light for indoor lettuce production is yet to be quantified. Lettuce ‘Cherokee’, ‘Green SaladBowl’, and ‘Little Gem’ were grown under 204 µmol·m^−2^·s^−1^ warm-white light-emitting diodes (LEDs) with supplemental far-red ranging from 5.3 to 75.9 µmol·m^−2^·s^−1^. Supplemental far-red light increased canopy light interception 5 days after the start of far-red light treatment (DAT) for ‘Green SaladBowl’ and ‘Little Gem’ and 7 DAT for ‘Cherokee’. The increase in light interception was no longer evident after 12 and 16 DAT for ‘Green SaladBowl’ and ‘Little Gem’, respectively. We did not find evidence that supplemental far-red light increased leaf-level photosynthesis. At the final harvest, shoot dry weights of ‘Cherokee’ and ‘Little Gem’ increased by 39.4% and 19.0%, respectively, while ‘Green SaladBowl’ was not affected. In conclusion, adding far-red light in indoor production increased light interception during early growth and likely increased whole plant photosynthesis thus growth, but those effects were cultivar-specific; the increase in dry weight was linear up to 75.9 µmol·m^−2^·s^−1^ far-red light.

## 1. Introduction

Depending on location and weather conditions, sunlight has about 19% far-red, defined here as light with wavelengths of 700–750 nm, relative to photosynthetically active radiation (PAR, 400–700 nm) [1]. Green leaves efficiently absorb UV light and PAR, with a small reduction in absorptance in the green region (400–500 nm). Leaves reflect and transmit more light in the far-red part of the spectrum [2,3]. Under plant canopies, therefore, light is relatively rich in far-red light and has a lower red/far-red ratio. Plants perceive a far-red enriched light environment as a signal of being shaded by other plant canopies. Responses of plants to shading can be classified into two categories: shade avoidance and shade tolerance. Shade-avoiding plants elongate more and branch less in response to a high far-red light environment, in an attempt to outgrow their neighbors and escape the shade [4]. Shade-tolerant plants acclimate to shade by increasing interception of available light and optimizing the light use efficiency of absorbed light, known as the ‘carbon gain hypothesis’ [5,6]. Shade-tolerant plants exhibit increased leaf expansion to increase light interception. They also tend to have low dark respiration, a low light compensation point, and a high quantum yield of CO_2_ assimilation, which together increase carbon gain per unit leaf area or unit soluble leaf protein and minimize maintenance carbon cost [5,6]. Plants in both categories share some common shade responses, such as increased specific leaf area (SLA, leaf area/shoot dry weight), reduced chlorophyll a:b ratio, and increased shoot:root ratio to boost light interception and utilization [4,5,6].

Regardless of avoidance or tolerance, shade responses of plants can be used to manipulate plant growth and morphology in controlled environment agriculture (CEA) production, where supplemental or sole-source lighting is used. A moderate shade avoidance response of plants improves light interception and efficiency of utilizing absorbed light [6]. CEA growers can enrich the light environment with far-red light to induce shade avoidance responses of crops, without reducing the light intensity and photosynthetic rate. For many vegetable and ornamental crops, such as lettuce, basil (*Ocimum basilicum*), coreopsis (*Coreopsis grandiflora*), geranium (*Pelargonium × hortorum*), pansy (*Viola × wittrockiana*), petunia (*Petunia × hybrida*), and snapdragon (*Antirrhinum majus*), inclusion of far-red light in CEA increases leaf area, canopy size and thus light interception, resulting in greater biomass [7,8,9,10,11,12,13,14].

In addition to morphological effects, far-red photons also have photosynthetic activity when combined with photons with shorter wavelengths. Traditionally, light with wavelengths of 400–700 nm is considered to be photosynthetically active and is designated as PAR. However, higher plants have two photosystems, photosystem I and photosystem II (PSI and PSII), that operate in a linear fashion to drive linear photosynthetic electron transport. PSII utilizes photons with wavelengths shorter than 685 nm [15,16]. Far-red photons have lower energy than PAR photons and cannot excite PSII. Thus, far-red photons have minimal photosynthetic activity when applied alone. PSI, on the other hand, can be excited by far-red photons. When applied along with PAR, far-red light can excite PSI and increase the quantum yield of photosystem II and photosynthesis [16,17]. Furthermore, light within the PAR region tends to over-excite PSII compared to PSI, with the exception of light between 570–620 nm and >690 nm, resulting in a sub-optimal quantum yield of oxygen evolution, a measure of PSII activity [17]. Since PSII and PSI work linearly, these two photosystems must work at matching rates to achieve high linear electron transport rates. Far-red photons can balance the excitation energy between the two photosystems and increase the quantum yield of PSII [16].

This concept is not unfamiliar to plant scientists. Over 60 years ago, far-red photons were shown to synergistically enhance photosynthesis when added to red photons, known as the ‘Emerson enhancement effect’ [18,19]. This synergistic effect has been long overlooked, partially due to influential work by McCree [2,20] defining PAR. In recent years, the addition of far-red light to PAR to improve quantum efficiency has gotten renewed attention. The synergistic photosynthetic enhancement from adding far-red light to PAR is well-documented for lettuce [1,16,21]. At the leaf level, adding far-red light to warm-white or red/blue light significantly increased the quantum yield of PSII (Φ_PSII_) and net photosynthetic rate (*P_n_*) of lettuce across a wide PAR range, from 50–750 µmol·m^−2^·s^−1^ [16]. At the canopy level, far-red photons, when combined with light of shorter wavelength, were shown to have equal photosynthetic efficiency as PAR photons [1,21]. This relationship held true across 14 different crops, including C3 plants and C4 species [1].

Another benefit of far-red inclusion in CEA lighting is the high efficacy (μmoles photon output per joule energy input) of far-red LEDs, which is greater than that of blue, red, and white LEDs [22]. Adding far-red LEDs to CEA has the potential to not only improve the crop’s light use efficiency by increasing photon capture and photosynthetic efficiency, but far-red LEDs also can reduce energy costs by replacing LEDs with a lower efficacy.

In summary, inclusion of far-red light in sole-source lighting can increase crop yield in two ways: (1) by inducing shade responses such as leaf expansion, which increases light interception by the plant canopy, and (2) by enhancing photosynthetic efficiency through the Emerson enhancement effect. Because far-red photons can have long-term effects on both plant morphology and physiology, the overall response of crop growth to far-red light is complicated.

A previous study on lettuce under sole-source lighting found that plants grown with additional far-red light had higher SLA, lower chlorophyll and nitrogen content per unit area, and subsequently lower leaf light absorptance and lower quantum yield of CO_2_ assimilation at *PPFD*s above 300 µmol·m^−2^·s^−1^ [11]. The negative effects of far-red light supplementation on leaf light absorptance and quantum yield of CO_2_ assimilation did not negate the beneficial effect of far-red light on growth, as both shoot fresh and dry weight increased under supplemental far-red light [11]. Another study similarly found that lettuce plants grown under a high far-red environment had greater biomass than lettuce plants adapted to a low far-red light environment at the same total *PFD* (integrated over wavelengths of 400 to 750 nm), despite lower chlorophyll content and leaf-level light absorptance [21]. At the whole plant level, lettuce plants in a far-red enriched environment had a similar quantum yield of CO_2_ assimilation as lettuce plants in a low far-red environment [21]. Although more far-red light can increase growth, excessive far-red light also can lead to undesirable stem elongation in lettuce [13]. Other undesirable traits were also observed in lettuce with far-red photon inclusion, such as reduced red coloration for red-leaf lettuce [10,11,23].

Although numerous studies showed the benefit of far-red photon inclusion in CEA light fixtures, there is a lack of studies that quantify the extent of benefit from far-red light inclusion and the optimal amount of far-red light to supply. We hypothesize that both the negative effect on leaf-level quantum yield of CO_2_ assimilation and the positive effect on canopy light interception of far-red light inclusion depend on far-red light intensity. This study aimed to examine the effect of a wide range of intensities of supplemental far-red light on lettuce plants in indoor production to identify the optimal *PFD* and the magnitude of benefit of supplemental far-red light, which would allow us to make recommendations for lettuce production under sole-source lighting. 

## 2. Results

Three lettuce cultivars, ‘Cherokee’, ‘Green Saladbowl’, and ‘Little Gem’, were grown in a walk-in growth chamber. All plants received warm white LED light (Appendix A) at 204 ± 11 µmol·m^−2^·s^−1^, which included 5.6 ± 0.3 µmol·m^−2^·s^−1^ far-red photons. Supplemental far-red light (peak at 724 nm, with a full width at half maximum of 18 nm) was provided ranging from 0 to 70.4 µmol·m^−2^·s^−1^, starting 6 days after seeding. Fifteen plants of each cultivar were grown in each of the 18 growing areas. We harvested seven plants at the first harvest (20 DAT) when canopies were closed, and another seven plants at the second harvest at 29 DAT (‘Green Saladbowl’) and 35 DAT (‘Cherokee’ and ‘Little Gem’) after the canopy had closed again.

### 2.1. Morphological Responses to Far-Red Light

At 7 DAT, far-red *PFD* linearly increased seedling height (Figure 1). Far-red photons increased ‘Cherokee’ height by 0.47 mm per µmol·m^−2^·s^−1^ of far-red photons (Figure 1A). Similarly, for both ‘Green Saladbowl’ and ‘Little Gem’, far-red light increased plant height by 0.22 and 0.75 mm per µmol·m^−2^·s^−1^ of far-red photons, respectively (Figure 1A). The relative increase in plant height was calculated as the ratio of increase in plant height induced by supplemental far-red light to average plant height without supplement far-red light (Figure 1B). The height of ‘Little Gem’ increased 229% when grown under the highest far-red light intensity compared to plants grown without supplemental far-red light, while the height of ‘Green Saladbowl’ increased by only 55% (Figure 1B). Similarly, at the first harvest at 20 DAT, far-red photons increased plant height of ‘Cherokee’, ‘Green Saladbowl’, and ‘Little Gem’ by 0.72, 0.71, and 1.25 mm per µmol·m^−2^·s^−1^ of far-red photons (Appendix A). Plant height of ‘Little Gem’ more than doubled under the highest far-red *PFD*, while the plant height of ‘Green Saladbowl’ and ‘Cherokee’ increased by about 70% under the highest far-red *PFD* (Appendix A).

Other morphological changes induced by far-red photons were evident at the first harvest as well. The length of the longest leaf increased linearly with increasing far-red *PFD* for all three cultivars (Figure 2), consistent with a shade acclimation response to a high far-red light environment. At the first harvest (20 DAT), far-red photons increased the length of the longest leaf of ‘Cherokee’, ‘Green Saladbowl’ and ‘Little Gem’ by 0.69, 1.15, and 1.12 mm per µmol·m^−2^·s^−1^, respectively (Figure 2A). There was also a significant interactive effect between far-red *PFD* and cultivar: the increase in longest leaf length with far-red *PFD* was larger for ‘Green Saladbowl’ and ‘Little Gem’ than for ‘Cherokee’ lettuce (*p* = 0.023). The relative increase in the longest leaf length was highest for ‘Little Gem’, which was 1.14% per µmol·m^−2^·s^−1^ of far-red photons, compared to 0.66% and 0.78% per µmol·m^−^^2^·s^−1^ increase in longest leaf length for ‘Cherokee’ and ‘Green Saladbowl’ (Figure 2B).

Canopy light interception was measured and also served as an indicator of canopy size. Light interception was calculated based on light measurements above and below the canopy. Canopy light interception increased with time in a sigmoidal pattern for all three cultivars. The effect of supplemental far-red photons on canopy light interception started to show at 7 DAT for ‘Cherokee’ and persisted until the first harvest (Figure 3 and Appendix A). The canopy light interception of ‘Green Saladbowl’ and ‘Little Gem’ was increased by far-red photons starting at 5 DAT (Appendix A), but this increase in light interception in response to far-red photons was no longer evident by 12 and 16 DAT for ‘Green Saladbowl’ and ‘Little Gem’, respectively (Appendix A). At 9 DAT (Figure 3), there was a positive correlation between light interception and far-red *PFD* for all three cultivars. At 9 DAT, ‘Cherokee’, ‘Green Saladbowl’ and ‘Little Gem’ lettuce grown without supplemental far-red intercepted 6.2, 8.8, and 4.8% photons of total incident photons, respectively (Figure 3A). With 70.4 µmol·m^−2^·s^−1^ of supplemental far-red photons, light interception of ‘Cherokee’, ‘Green Saladbowl’, and ‘Little Gem’ increased to 12.8, 16.5%, and 12.5%, respectively (Figure 3A). Although light interception was low, the relative increase in light interception of ‘Cherokee’, ‘Green Saladbowl’ and ‘Little Gem’ plants grown under the highest far-red *PFD* were 120, 78, and 153%, respectively, compared to plants grown without supplemental far-red light (Figure 3B).

Lettuce cultivar affected light interception (*p* < 0.01), but there was no interactive effect of cultivar and far-red *PFD* on light interception (*p* = 0.52), indicating that light interception of all three cultivars changed in a similar manner with far-red *PFD* (e.g., similar slopes in Figure 3A). Light interception of ‘Green Saladbowl’ lettuce increased faster over time than light interception of ‘Cherokee’ and ‘Little Gem’ (Appendix A) during the seedling stage. Starting at 7 DAT, light interception of ‘Green Saladbowl’ was higher than light interception of ‘Cherokee’ and ‘Little Gem’, averaged across different far-red *PFD*s (Appendix A). At 12 DAT, light interception of ‘Cherokee’ lettuce caught up with ‘Green Saladbowl’, and light interception of ‘Green Saladbowl’ was no longer the highest among the three cultivars (Appendix A). The light interception of ‘Little Gem’ was consistently lowest among the three cultivars (Appendix A).

Leaf area at the first harvest also increased in response to supplemental far-red *PFD* in ‘Cherokee’ and ‘Little Gem’, but not in ‘Green Saladbowl’ (Figure 4). ‘Cherokee’ and ‘Little Gem’ plants under the highest far-red *PFD* had 131 and 166 cm^2^·plant^−1^ greater leaf area than plants grown without supplemental far-red photons (Figure 4A), which represented a 52.1% and 38.0% increase in leaf area, compared to plants without supplemental far-red light (Figure 4B). At the second harvest, leaf area of ‘Green Saladbowl’ increased with higher far-red *PFD*, but this was not the case for ‘Cherokee’ and ‘Little Gem’ (Figure 5). 70.4 µmol·m^−2^·s^−1^ supplemental far-red *PFD* resulted in 333 cm^2^·plant^−1^ larger leaf area for ‘Green Saladbowl’ (Figure 5A), which represented a 20.8% increase compared to plants without supplemental far-red light (Figure 5B).

We did not find a consistent trend in specific leaf weight (shoot dry weight divided by total leaf area) at the two harvests. At the first harvest, specific leaf weight decreased with far-red *PFD* for ‘Little Gem’, but not for ‘Cherokee’ and ‘Green Saladbowl’ (Appendix A). For ‘Green Saladbowl’, the correlation between far-red *PFD* and specific leaf weight was borderline significant (*p* = 0.075). Contrary to results from the first harvest, specific leaf weight of ‘Cherokee’ increased with increasing far-red *PFD* at the second harvest (Appendix A). No effect of far-red *PFD* on specific leaf weight of ‘Little Gem’ and ‘Green Saladbowl’ was observed at the second harvest (Appendix A).

Leaf pigments were also affected by the addition of far-red photons (Figure 6 and Figure 7). Leaf chlorophyll content index (CCI) and anthocyanin content index (ACI) were measured twice before the first harvest: at 12 (data not shown) and at 16 DAT (Figure 6 and Figure 7). Far-red photons decreased CCI for all three lettuce cultivars at both dates. The ACI was only measured for ‘Cherokee’, a red lettuce cultivar, and was negatively correlated with far-red *PFD* at 16 DAT (Figure 7). No effect of far-red photons on ACI was detected at 12 DAT (data not shown).

Excessive stem elongation of lettuce is undesirable. At the second harvest, we observed elongated stems for ‘Cherokee’ and ‘Green Saladbowl’ under supplemental far-red photons, but not for ‘Little Gem’ (Figure 8). Stems of ‘Cherokee’ and ‘Green Saladbowl’ lettuce under the highest far-red *PFD* were 97% and 179% longer than those without supplemental far-red light (Figure 8B). However, since the stems were short to start with, the stem length of ‘Cherokee’ and ‘Green Saladbowl’ increased by only 1.0 and 1.8 cm (Figure 8A). These increases in stem length were statistically significant, but not horticulturally meaningful.

### 2.2. Leaf-Level Photosynthetic Response to Supplemental Far-Red Light

We found no significant effect of far-red *PFD* on Φ_PSII_ or *P_n_* of all three cultivars. The Φ_PSII_ at 27 DAT of ‘Cherokee’, ‘Green Saladbowl’ and ‘Little Gem’ lettuce averaged 0.706, 0.682, and 0.682, respectively, and was not significantly affected by far-red *PFD* (Figure 9). Similarly, the leaf level *P_n_* of all three cultivars was not affected by far-red *PFD*. The *P_n_* of ‘Cherokee’, ‘Green Saladbowl’, and ‘Little Gem’ at 28 DAT averaged 6.25, 5.18, and 7.25 µmol·m^−2^·s^−1^, respectively, regardless of far-red *PFD* (data not shown).

In addition, we measured leaf-level Φ_PSII_ and *P_n_* under their growing spectrum, then turned off the supplemental far-red light and measured Φ_PSII_ and *P_n_* under white light only at 28 DAT. The effect of supplemental far-red photons on electron transport and leaf photosynthesis was expressed as the ratio of Φ_PSII_ or *P_n_* under white light with supplemental far-red light to Φ_PSII_ or *P_n_* under white LED light only (Figure 10 and Appendix A), referred to as Φ_PSII,FRon_ to Φ_PSII,FRoff_ ratio and *P*_net,FRon_ to *P*_net,FRoff_ ratio. We did not find any significant differences in Φ_PSII_ and *P_n_* under white LED light with and without supplemental far-red light (Figure 10).

### 2.3. Final Yield

Shoot dry weight of ‘Cherokee’ and ‘Little Gem’ was positively correlated with far-red *PFD* at both 20 DAT and 35 DAT, but the dry weight of ‘Green SaladBowl’ was not affected at either harvest (Figure 11 and Figure 12). At the first harvest, the highest supplemental far-red PFD (70.4 µmol·m^−2^·s^−1^) resulted in 0.264 g·plant^−1^ (38.1%) and 0.267 g·plant^−1^ (31.0%) higher shoot dry weight for ‘Cherokee’ and ‘Little Gem’ (Figure 11), compared to plants without supplemental far-red. At the second harvest, shoot dry weights of ‘Cherokee’ and ‘Little Gem’ plants were similar without supplemental far-red light (3.81 and 3.92 g·plant^−1^, respectively) (Figure 12A), but responded differently to far-red *PFD*. Supplemental far-red light increased the shoot dry weight of ‘Cherokee’ and ‘Little Gem’ by 0.023 g and 0.011 g per plant per µmol·m^−2^·s^−1^ far-red photons (Figure 12A). The relative increase in shoot dry weight in response to 70.4 µmol·m^−2^·s^−1^ supplemental far-red *PFD* was thus higher for ‘Cherokee’ (39.4%) than for ‘Little Gem’ (19.0%) (Figure 12B).

For ‘Little Gem’ and ‘Green Saladbowl’, the effect of far-red *PFD* on total shoot fresh weight was not statistically significant at either harvest (Appendix A). At the 1st harvest, shoot fresh weight of ‘Cherokee’ lettuce increased by 6.94 g·plant^−1^ with the highest supplemental far-red *PFD* (Appendix A), representing a 38.0% increase in shoot fresh weight compared to plants grown without supplemental far-red light (Appendix A). At the 2nd harvest, the highest (70.4 µmol·m^−2^·s^−1^) supplemental far-red *PFD* resulted in 27.5 g·plant^−1^ more shoot fresh weight compared to plants without supplemental far-red photons for ‘Cherokee’ (Appendix A), a 26.8% increase in shoot fresh weight compared to plants without far-red light (Appendix A).

## 3. Discussion

### 3.1. Morphological Changes Induced by Adding Far-Red Light

Adding far-red light resulted in longer leaves (Figure 2) and longer stems (Figure 8), as well as an upward-growing habit (hyponasty) (Figure 1 and Appendix A), which is consistent with shade responses [4]. The mechanism of plants’ morphological responses to far-red light is well understood. Red light and far-red light shift the balance between two photo-reversible isoforms of phytochrome: red photons convert Pr to Pfr and far-red photons convert Pfr to Pr. Changes in the phytochrome photo-equilibrium (PPE; ratio of Pfr to total phytochrome) induce downstream morphological responses [4,5,6,9,24]. A low PPE induces shade response in lettuce, such as leaf expansion, stem elongation, and hyponasty [24].

A positive correlation between leaf expansion and supplemental far-red *PFD* was observed for all three cultivars (Figure 2). Leaf expansion as a result of supplemental far-red light was previously observed in various lettuce cultivars [9,12,23], but the extent of leaf expansion varied among cultivars. A 1.1 mm per µmol·m^−2^·s^−1^ increase in leaf length in response to supplemental far-red *PFD* was previously observed for ‘Green Saladbowl’ under white LED light of similar *PPFD* [12], similar to the 1.12 mm per µmol·m^−2^·s^−1^ increase in leaf length observed in our study (Figure 2).

Supplemental far-red *PFD* linearly increased plant height in the seedling stage (Figure 1) and after canopy closure at the first harvest (Appendix A). This was the combined result of longer leaf length and hyponasty; leaves under higher far-red *PFD* grew at a steeper, upward angle. This is consistent with a shade avoidance response [4,6]. Hyponasty, resulting from increased far-red *PFD*, can create a more open canopy structure, allowing for a more uniform light distribution throughout the canopy, which in turn increases canopy photosynthesis [11].

Greater total leaf area, because of leaf expansion induced by supplemental far-red, was observed for ‘Cherokee’ and ‘Little Gem’, but not for ‘Green Saladbowl’ at the first harvest (Figure 4). Supplemental far-red photons did promote leaf expansion of ‘Green Saladbowl’ lettuce at an early stage, as is evident from the higher light interception from 5 DAT to 9 DAT induced by supplemental far-red photons (Figure 3, Appendix A) and larger longest leaf length at 20 DAT (Figure 2). However, later on, the rapid growth of ‘Green Saladbowl’ lettuce created a denser canopy and more self-shading in the canopy, compared to ‘Cherokee’ and ‘Little Gem’ lettuce, which may have diminished the effect of supplemental far-red *PFD* on total leaf area. ‘Green Saladbowl’ lettuce had the highest light interception among the three lettuce cultivars from 5 DAT to 9 DAT (Figure 3, Appendix A). It can also be seen at the 1st harvest that ‘Green Saladbowl’ lettuce had a higher total leaf area than ‘Cherokee’ and ‘Little Gem’ lettuce (*p* = 0.02) (Figure 4A), which suggests more leaf overlap and self-shading. The stronger self-shading might mask the effect of supplemental far-red photons on leaf area of ‘Green Saladbowl’ at the 1st harvest.

A previous study found that supplemental far-red photons increased lettuce biomass production mainly through leaf expansion and higher light interception. The effect of supplemental far-red photons on lettuce was stronger at low planting density than at high planting density [25]. They argued that for lettuce at high planting density, the light interception was already high; thus, supplemental far-red photons had a limited effect on plant growth [25]. This theory applies to our study as well. It is possible that light interception was already high and the red:far-red ratio was low within the dense canopy of ‘Green Saladbowl’ lettuce after the early growth stage. Thus, supplemental far-red photons did not increase leaf area of ‘Green Saladbowl’ but increased the leaf area of ‘Cherokee’ and ‘Little Gem’ at the first harvest (Figure 4). After the 1st harvest, when eight of the 15 plants were removed, ‘Green Saladbowl’ plants quickly closed the canopy gaps created by the harvest. Therefore ‘Green Saladbowl’ lettuce was harvested nine days after the 1st harvest, whereas ‘Cherokee’ and ‘Little Gem’ were harvested 15 days after the 1st harvest. At this time, supplemental far-red photons increased leaf area of ‘Green Saladbowl’, but did not increase leaf area of ‘Cherokee’ and ‘Little Gem’ (Figure 5).

Chlorophyll and anthocyanin biosynthesis are affected by PPE [9,26,27]. In lettuce, lower chlorophyll and lower anthocyanin for red-leaf cultivars were previously observed under higher far-red *PFD* [9,10,11]. In our study, CCI of all three cultivars decreased with increasing far-red *PFD* (Figure 6). ACI of the red-leaf ‘Cherokee’ also decreased with increasing far-red *PFD* (Figure 7). These are likely results of reduced pigment synthesis under high far-red *PFD*, rather than lower specific leaf weight (SLW) (as seen in Appendix A), as we found no correlation between CCI and SLW at the 1st harvest (*p* = 0.86 for ‘Cherokee’, *p* = 0.07 for ‘Green Saladbowl’ and *p* = 0.30 for ‘Little Gem’). For the red cultivar ‘Cherokee’, ACI also was not correlated with SLW (*p* = 0.84). Previous studies found that light with a high red:far-red ratio converts the PPE towards Pfr, which stimulates both chlorophyll synthesis and anthocyanin synthesis [26,28,29]. Our results agree with the previous observation that the reduction in chlorophyll and anthocyanin content was a result of decreased biosynthesis rather than decreased SLW.

Other parameters related to crop quality were also affected by supplemental far-red *PFD*. Stem length (Figure 8) and internode length (data not shown) of ‘Cherokee’ and ‘Green Saladbowl’ increased linearly with increasing far-red *PFD*. Stem elongation was previously also observed for other lettuce cultivars [12,13]. In our study, ‘Cherokee’ and ‘Green Saladbowl’ plants grown under the highest far-red *PFD* had an 84.5% and 187.9% increase, respectively, in stem length compared to plants without supplemental far-red light (Figure 8B). However, the absolute increase was small, only 1.0 and 1.8 cm for ‘Cherokee’ and ‘Green Saladbowl’, respectively. The supplemental far-red light did not affect stem length of ‘Little Gem’ plants (Figure 8). A shade avoidance response may reduce the productivity of field crops, as plants allocate more resources to the stem, in competition with leaf and root [4]. However, in our study, with up to 70.4 µmol·m^−2^·s^−1^ supplemental far-red light added to 204 µmol·m^−2^·s^−1^ white light, the length of lettuce stems was still small.

### 3.2. No Evidence That Far-Red Photons Affected Leaf-Level Photosynthesis

We found no difference in leaf-level *P_n_* and Φ_PSII_ among plants grown under different supplemental far-red *PFDs* (Figure 9). A previous study found that lettuce plants that grew in a far-red supplemented environment had similar Φ_PSII_ but lower *P_n_*, than plants under only white LED light, particularly at high *PPFD* [11]. Lettuce leaves from a far-red supplemented environment had lower leaf nitrogen concentration per unit area, chlorophyll concentration per unit area, light absorptance, and stomatal conductance, which resulted in lower *P_n_* at high *PPFD* [11]. In our study, lettuce leaves responded to supplemental far-red *PFD* by lowering CCI (Figure 6). Lower CCI correlates with lower chlorophyll concentration in leaves and could lead to lower leaf-level light absorptance [30,31,32]. It may explain our observation that the leaf-level *P_n_* and Φ_PSII_ was not increased by additional far-red photons (Figure 9).

Leaf-level *P_n_* and Φ_PSII_ did not significantly change either when supplemental far-red light was turned off (Figure 10 and Appendix A). Far-red photons have been shown to efficiently drive photosynthesis of lettuce at both leaf and canopy level [1,11,16,21]. Previous studies also showed that temporarily turning on or off far-red lights affected both leaf level and canopy level photosynthetic rates of lettuce whether developed under supplemental far-red light or not [1,11,21]. In our study, however, we did not find any changes in *P_n_* and Φ_PSII_ of lettuce leaves when turning off supplemental far-red lights. This is likely because our photosynthetic measurements with and without far-red light were not taken on the same spot of leaves. The cuvette of the leaf gas exchange system was removed and re-clamped on the leaves before and after far-red lights were turned off. Therefore, our data may have failed to capture the change in leaf level *P_n_* and Φ_PSII_ after far-red lights were turned off. In the case of ‘Cherokee’, a red-leaf lettuce cultivar, the ACI decreased linearly with increasing far-red *PFD* (Figure 7). Anthocyanins may competitively absorb light energy and divert absorbed energy away from photosynthesis [33]. Nevertheless, in our study, the anthocyanins in ‘Cherokee’ leaves had no significant effect on leaf-level *P_n_* (*p* = 0.53). In summary, we did not find evidence that supplemental far-red photons increased leaf photosynthesis, likely due to noise in our photosynthetic data.

### 3.3. Supplemental Far-Red Light Increased Final Yield in a Cultivar-Specific Manner

Shoot dry weight of ‘Cherokee’ and ‘Little Gem’, but not that of ‘Green Saladbowl’, increased with increasing far-red *PFD* at both harvests (Figure 11 and Figure 12). Higher light interception, resulting from increased leaf elongation in response to supplemental far-red light, increased the amount of light energy available for photosynthesis and would thus be expected to increase whole-canopy photosynthetic rates and accelerate plant growth. This would be especially important for young plants, since their canopy size and light interception are small. Higher light interception in response to supplemental far-red photons during the seedling stage increases seedlings’ light energy capture and carbon assimilation, thus promoting canopy enlargement and light capture.

In our study, supplemental far-red light increased the final dry weight of “Little Gem’ and ‘Cherokee’ lettuce through increasing light interception, but not that of ‘Green Saladbowl’. The importance of light interception for crop growth is evident from the correlations between light interception and shoot dry weight. Shoot dry weight at the 1st harvest correlated with light interception of ‘Cherokee’ (at 12 and 16 DAT), ‘Green Saladbowl’ (12 to 19 DAT), and ‘Little Gem’ (9 DAT till the 1st harvest). Light interception starting from 7 DAT until the 1st harvest also strongly correlated with the final shoot dry weight of all three cultivars at the 2nd harvest, except for the light interception of ‘Green Saladbowl’ at 9 DAT. For example, light interception by ‘Cherokee’, ‘Green Saladbowl’, and ‘Little Gem’ at 12 DAT was strongly correlated with shoot dry weight at both the 1st and 2nd harvest (Figure 13). The dry weight at the 1st harvest did not correlate with the quantum yield of photosystem II for any of the cultivars (*p* ≥ 0.44, *r* ≤ 0.19). Leaf-level photosynthesis often does not correlate with crop yield since it is a short-term measurement, limited to a small section of a single leaf and thus not representative of the entire canopy during the growing cycle [34]. These results suggest that the changes in dry weight induced by supplemental far-red *PFD* were due mainly to changes in light interception rather than leaf-level photosynthesis.

Far-red inclusion has been shown to increase canopy size and light interception, thus resulting in greater biomass for a wide range of vegetable and ornamental crops, including lettuce [7,8,9,10,11,12,13,14]. Throughout this study, ‘Cherokee’ plants grown under high far-red *PFD* consistently intercepted more light than plants grown under low far-red *PFD*, which resulted in higher shoot dry weight and fresh weight at both harvests (Figure 11, Figure 12, Appendix A). With the inclusion of far-red photons, plants enter a ‘self-re-enforcing’ cycle: larger leaves and better light interception boost plant growth, further enhancing light interception and resulting in higher dry weight [12]. Light interception of ‘Green Saladbowl’ plants was increased by supplemental far-red photons only from 5 to 9 DAT (Figure 3 and Appendix A). It is possible that the rapid growth of ‘Green Saladbowl’ lettuce created self-shading within their canopies, which masked the effect of supplemental far-red photons. In contrast, for ‘Little Gem’, where supplemental far-red *PFD* only increased light interception during a short window of the seedling stage (5–12 DAT) (Figure 3 and Appendix A), the benefit of the increased light interception nevertheless carried over to final dry weight (Figure 11 and Figure 12). In short, this self-re-enforcing cycle appears to be cultivar dependent.

Previous research suggested that plants grown with supplemental far-red light had an open canopy structure, facilitating more uniform light distribution within the canopy, and increased light use efficiency and whole canopy photosynthesis [11]. We observed a similar change in canopy structure as a consequence of change in leaf angle (hyponasty) and taller plants (Figure 1 and Appendix A). It is possible that better light penetration into canopy similarly increased canopy-level photosynthesis in our study. Better light penetration possibly benefits the growth of ‘Cherokee’ and ‘Little Gem’ lettuces more than ‘Green Saladbowl’ lettuces, as ‘Green Saladbowl’ is a loose-leaf lettuce cultivar, which lacks a tight canopy architecture.

### 3.4. Conclusions

Increasing far-red *PFD* linearly increased leaf expansion with a background of 204 µmol·m^−2^·s^−1^ white LED light. The increase in light interception resulting from this leaf expansion was cultivar dependent. Supplemental far-red photons increased light interception of ‘Cherokee’ throughout the whole production cycle and resulted in higher shoot fresh and dry weight. The increase in light interception caused by supplemental far-red photons was transient for ‘Green Saladbowl’ and ‘Little Gem’, and the transient benefits of increased light interception translated to higher shoot dry weight for ‘Little Gem’, but not for ‘Green Saladbowl’. Supplemental far-red light resulted in lower chlorophyll content for all three cultivars and lower anthocyanin content for the red lettuce ‘Cherokee’. Those changes in pigmentation did not significantly affect Φ_PSII_. In summary, adding far-red light in indoor production increased light interception of lettuce seedlings, but did not significantly affect leaf photosynthesis, and resulted in linear increases in biomass, up to 70.4 µmol·m^−2^·s^−1^ in two of the three cultivars tested.

## 4. Materials and Methods

### 4.1. Walk-In Cooler Setup

Three metal shelves were installed inside a walk-in growth chamber. Each shelving unit had 3 levels, and each level was divided into two 1.2 × 0.6 m units, resulting in 18 separate growing areas. During the study, the temperature in the growth chamber was 20.4 ± 0.2 °C (average ± standard deviation), vapor pressure deficit was 0.74 ± 0.13 kPa, and the CO_2_ concentration was 819 ± 44 µmol·mol^−1^.

### 4.2. Plant Material

Three lettuce cultivars were used in this study: ‘Green Saladbowl’, ‘Little Gem”, and ‘Cherokee’ (Johnny’s seeds, Winslow, ME, USA). Plants were grown from seed in 10-cm square pots filled with a peat-based soilless substrate (Fafard 4P, Sun Gro Horticulture, Agawam, MA, USA). One tray with 15 plants of each cultivar was placed in each section of the shelving units, totaling 54 trays of lettuce, 18 for each cultivar. Plants were sub-irrigated with nutrient solution once a day. The nutrient solution contained 100 mg·L^−1^ N (Peters Excel 15-5-15 Cal-Mag water-soluble fertilizer; 15N-2.2P-12.5K, Everris, Dublin, OH, USA). During germination, trays on the metal shelves were rotated periodically within the same section to minimize the impact of any environmental gradients.

### 4.3. Far-Red Light Treatments

Three trays, one for each of the three cultivars, were placed in each one of the 18 growing areas. Seeds were germinated under white LED light (Appendix A) of 204 ± 11.0 µmol·m^−2^·s^−1^ with a 16 h photoperiod. After germination (6 days after sowing), far-red LED lights were turned on. Plants in three of the 18 units were grown under white LED light only as a control, while plants in the remaining 15 units were grown with supplemental far-red *PFD*s ranging from 0.0 to 70.4 µmol·m^−2^·s^−1^ (Appendix A). The white LED light contained about 5.3 µmol·m^−2^·s^−1^ far-red light, the total far-red *PFD* plants received ranged from 5.3 to 75.9 µmol·m^−2^·s^−1^. Far-red light had the same photoperiod as the white LED light.

### 4.4. Physiological and Morphological Measurements

Light interception of the crops was measured using a ceptometer (ACCUPAR LP-80, METER Group, Pullman, WA, USA). The reference light intensity was measured at the substrate level before germination. After plants emerged, under-canopy light intensity was measured similarly at the substrate level twice a week. Light interception by plants was then calculated as the relative reduction of the light at the substrate level to the reference light intensity. Plant height was measured for three lettuce plants per experimental unit at 7 and 20 DAT (at first harvest). Chlorophyll content index (CCI) and anthocyanin content index (ACI) were measured with a CCM-200+ chlorophyll content meter and ACM-200+ anthocyanin content meter (Opti-Sciences, Hudson, NH, USA), respectively, at 12 DAT and 16 DAT.

At 27 DAT, Φ_PSII_ of a newly fully expanded leaf of each cultivar was measured under the light environment plants were grown under using a fluorometer (Junior-PAM, Heinz Walz GmbH, Effeltrich, Germany). At 28 DAT, *P_n_* and Φ_PSII_ of a recently fully expanded leaf were measured on one plant of each cultivar in 11 of the 18 experimental units, using the same fluorometer and a leaf gas exchange system (CIRAS-3, PP system, Amesbury, MA, USA). We then turned off the supplemental far-red light, waited for 20 min, and measured *P_n_* and Φ_PSII_ on the same leaf (but not the same location) under only white LED light to quantify the effect of supplemental far-red light on photochemistry and photosynthesis. All *P_n_* and Φ_PSII_ measurements were taken under a *PPFD* of about 200 µmol·m^−2^·s^−1^.

The first harvest occurred at 20 DAT. Seven plants from each experimental unit were harvested at the first harvest. During the first harvest, canopy height and length of the longest leaf from three plants of each experimental unit were measured. Total leaf area, shoot fresh weight, and shoot dry weight were measured on all seven plants combined. The second harvest of ‘Cherokee’ and ‘Little Gem’ occurred at 35 DAT and that of ‘Green Saladbowl’ at 29 DAT. ‘Green Saladbowl’ was harvested earlier due to its rapid growth. During the second harvest, stem length of three plants in each experimental unit was measured in addition total leaf area, shoot fresh weight, and shoot dry weight of seven plants. Total leaf area was measured using a leaf area meter (Li-3100, Li-Cor Biosciences, Lincoln, NE, USA). The dry weight was measured after shoot tissue was dried at 80 °C for at least 72 h. Specific leaf weight was calculated as shoot dry weight divided by total leaf area. One extra plant was discarded before canopy closure after the 1st harvest.

### 4.5. Data Analysis

This study used a completely randomized design with a split-plot for the three cultivars, with far-red *PFD* randomly assigned to the 18 growing spaces. Data were analyzed using regression analysis in Microsoft Excel (Microsoft, Seattle, WA, USA) to detect the effect of far-red *PFD* on physiological and morphological parameters. Correlations were tested with multivariate analysis of JMP Pro 15 to detect correlations between leaf area and light interception, specific leaf area, and CCI, as well as light interception and lettuce dry weight (SAS Institute, Cary, NC, USA). ANOVA was also performed in JMP Pro 15.

## Figures and Tables

**Figure 1 plants-11-02714-f001:**
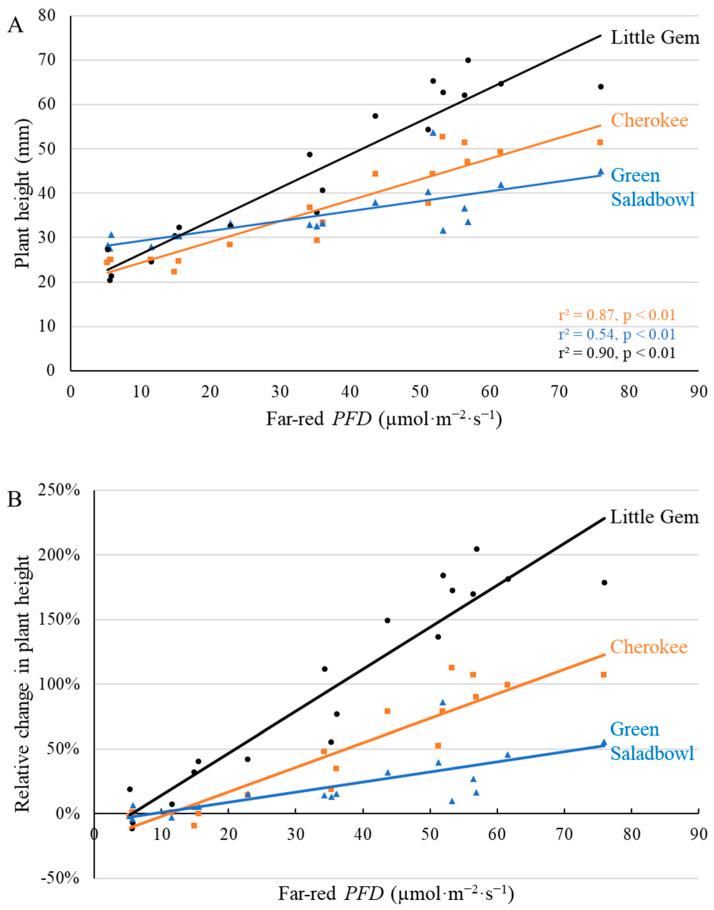
With a background of 204 µmol·m^−2^·s^−1^ white LED light, supplemental far-red photons linearly increased plant height (**A**) and the relative change in plant height ((**B**), compared to plants under no supplemental far-red light) 7 days after the start of far-red light treatment for ‘Cherokee’, ‘Green Saladbowl’, and ‘Little Gem’ lettuce. The *r*^2^ and *p* values for regression of each cultivar apply to both graphs and are shown with the respective color.

**Figure 2 plants-11-02714-f002:**
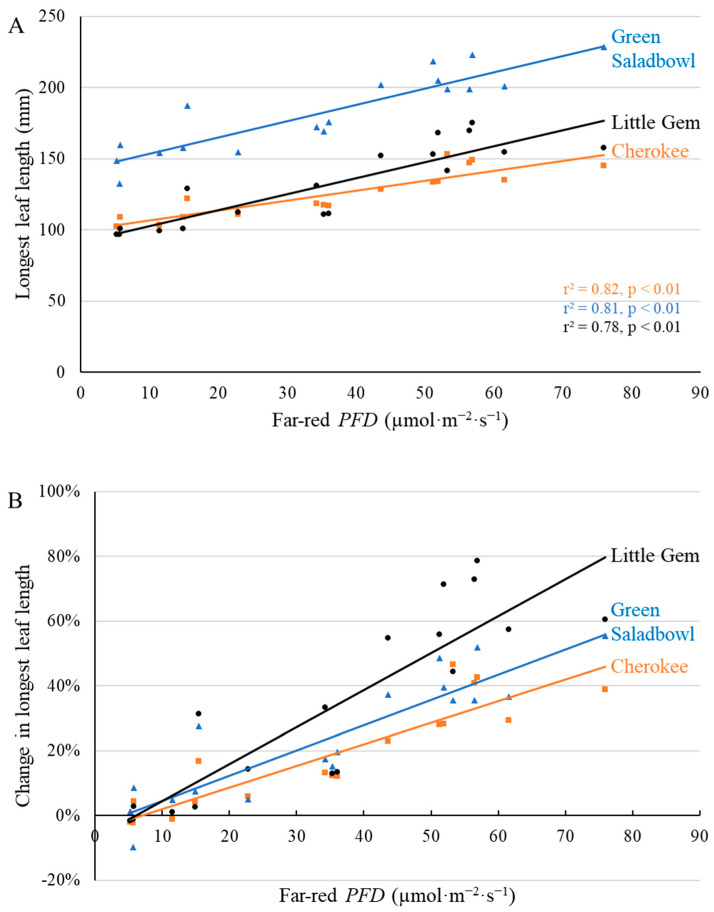
With a background of 204 µmol·m^−2^·s^−1^ white LED light, supplemental far-red photons linearly increased length of the longest leaf (**A**) and the relative change in length of the longest leaf ((**B**), compared to plants under no supplemental far-red light) 20 days after the start of far-red light treatment for ‘Cherokee’, ‘Green Saladbowl’, and ‘Little Gem’ lettuce. The *r*^2^ and *p* values for regression of each cultivar apply to both graphs and are shown with the respective color.

**Figure 3 plants-11-02714-f003:**
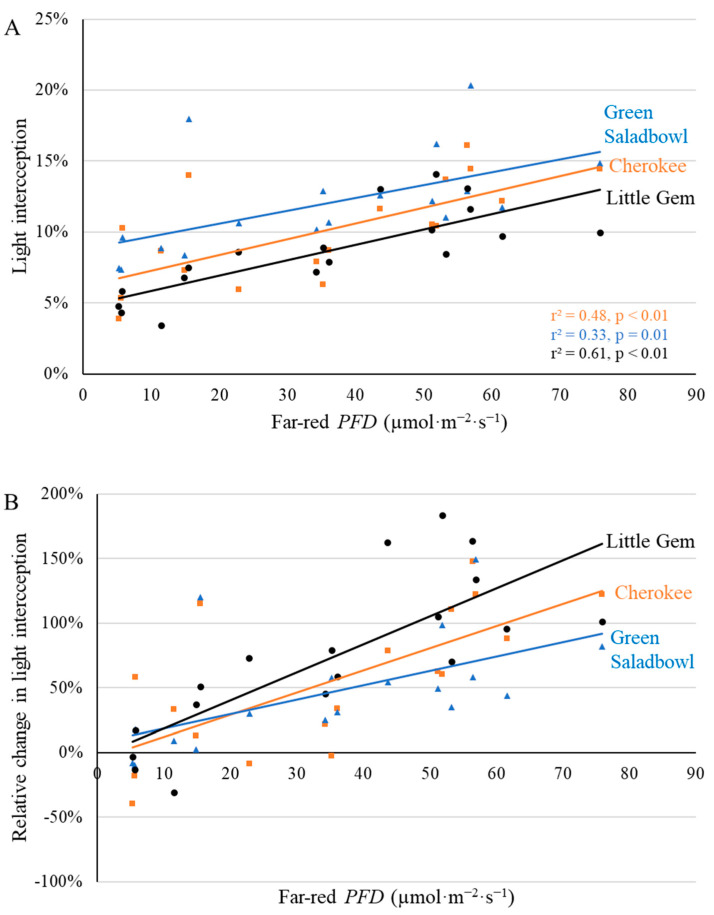
With a background of 204 µmol·m^−2^·s^−1^ white LED light, supplemental far-red photons linearly increased canopy light interception (**A**) and the relative change in canopy light interception ((**B**), compared to plants under no supplemental far-red light) 9 days after the start of far-red light treatment for ‘Cherokee’, ‘Green Saladbowl’, and ‘Little Gem’ lettuce. The *r*^2^ and *p* values for regression of each cultivar apply to both graphs and are shown with the respective color.

**Figure 4 plants-11-02714-f004:**
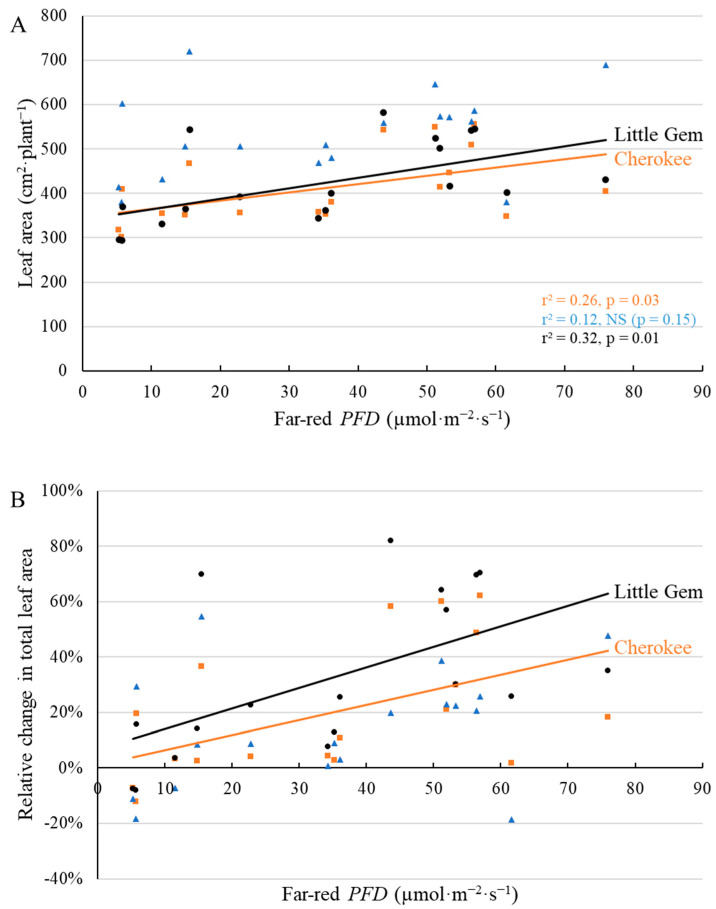
With a background of 204 µmol·m^−2^·s^−1^ white LED light, supplemental far-red photons linearly increased leaf area (**A**) and the relative change in leaf area ((**B**), compared to plants without supplemental far-red light) at 21 days after the start of far-red light treatment for ‘Cherokee’ and ‘Little Gem’ lettuce, but not for ‘Green Saladbowl’ (blue triangles). The *r*^2^ and *p* values for regression of each cultivar apply to both graphs and are shown with the respective color.

**Figure 5 plants-11-02714-f005:**
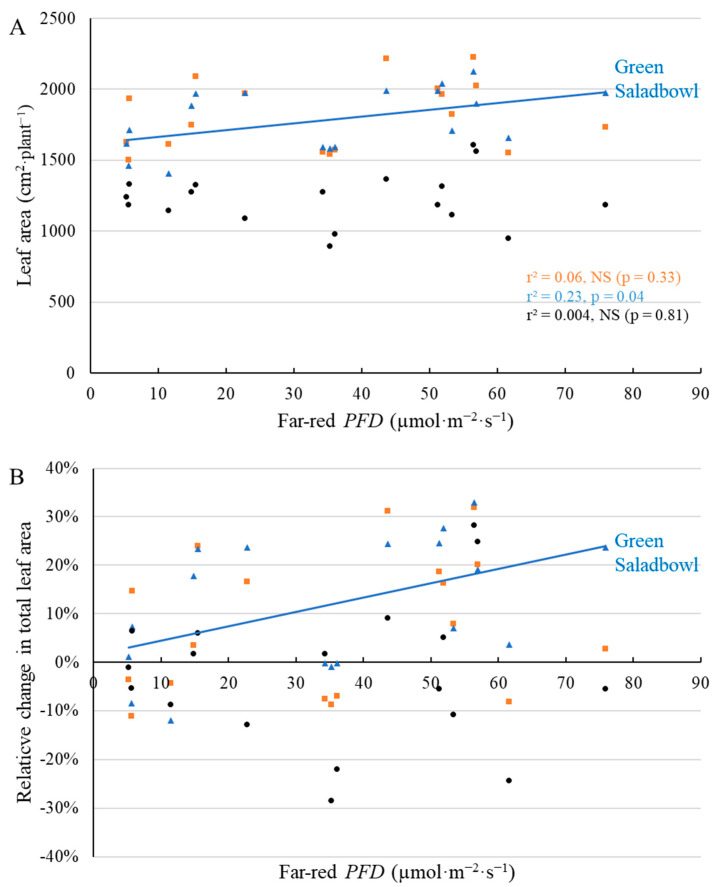
With a background of 204 µmol·m^−2^·s^−1^ white LED light, supplemental far-red photons linearly increased total leaf area (**A**) and the relative change in total leaf area ((**B**), compared to plants without supplemental far-red light) 35 days after the start of far-red light treatment for ‘Green Saladbowl’ lettuce, but not for ‘Cherokee’ (orange squares) and ‘Little Gem’ (black dots). The *r*^2^ and *p* values for regression of each cultivar apply to both graphs and are shown with the respective color.

**Figure 6 plants-11-02714-f006:**
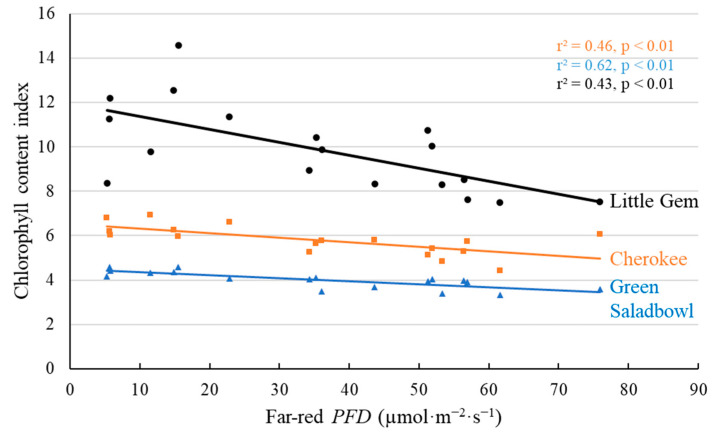
With a background of 204 µmol·m^−2^·s^−1^ white LED light, supplemental far-red photons linearly decreased chlorophyll content index at 16 days after the start of far-red light treatment for ‘Cherokee’, ‘Green Saladbowl’ and ‘Little Gem’ lettuce. The *r*^2^ and *p* values for regression of each cultivar are shown with the respective color.

**Figure 7 plants-11-02714-f007:**
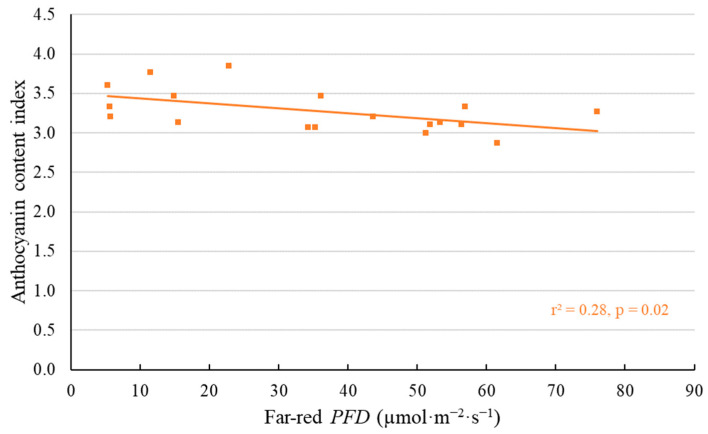
With a background of 204 µmol·m^−2^·s^−1^ white LED light, supplemental far-red photons linearly decreased anthocyanin content index of ‘Cherokee’ lettuce at 16 days after the start of far-red light treatment. The *r*^2^ and *p* value are shown in the graph.

**Figure 8 plants-11-02714-f008:**
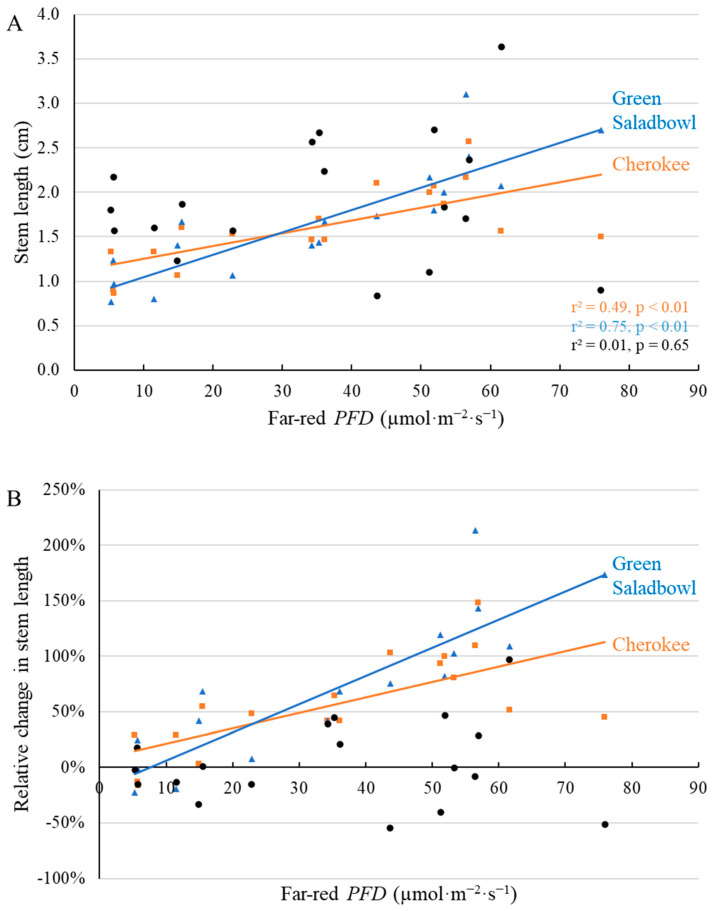
With a background of 204 µmol·m^−2^·s^−1^ white LED light, supplemental far-red photons linearly increased stem length (**A**) and relative stem length ((**B**), compared to plants under no supplemental far-red light) 35 days after the start of far-red light treatment for ‘Cherokee’ and ‘Green Saladbowl’, but not for ‘Little Gem’ (black dots) lettuce. The *r*^2^ and *p* values for regression of each cultivar apply to both graphs and are shown with the respective color.

**Figure 9 plants-11-02714-f009:**
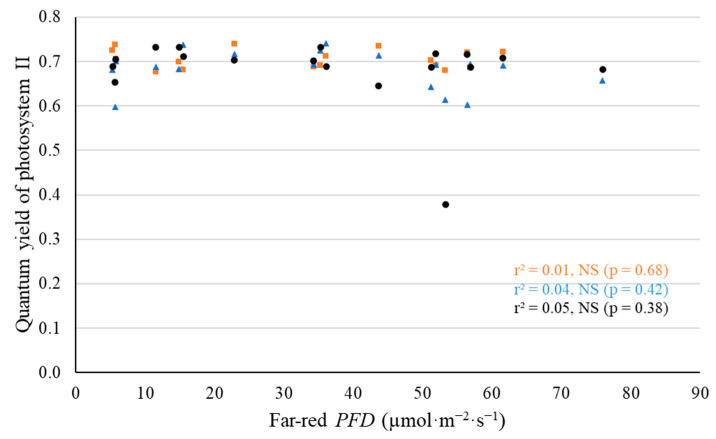
With a background of 204 µmol·m^−2^·s^−1^ white LED light, supplemental far-red photons had no effect on quantum yield of photosystem II at 27 days after the start of far-red light treatment for ‘Cherokee’ (orange squares), ‘Green Saladbowl’ (blue triangles), and ‘Little Gem’ (black dots) lettuce.

**Figure 10 plants-11-02714-f010:**
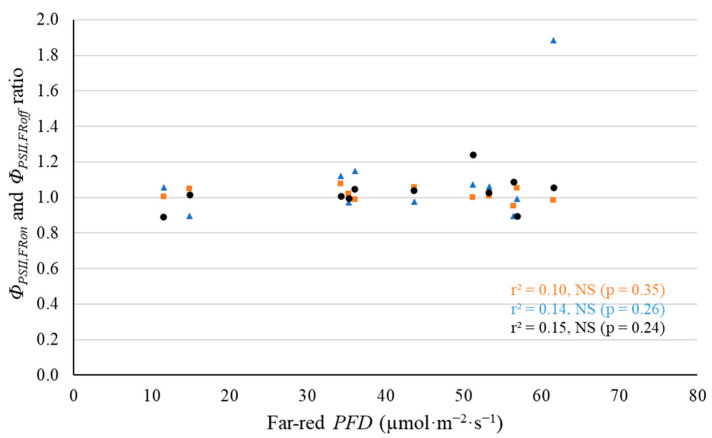
With a background of 204 µmol·m^−2^·s^−1^ white LED light, turning off supplemental far-red lights had no effect on quantum yield of photosystem II for ‘Cherokee’ (orange squares), ‘Green Saladbowl’ (blue triangles) and ‘Little Gem’ (black dots) lettuce 28 days after the start of far-red light treatment. The *r*^2^ and *p* values for regression of each cultivar are shown with the respective color.

**Figure 11 plants-11-02714-f011:**
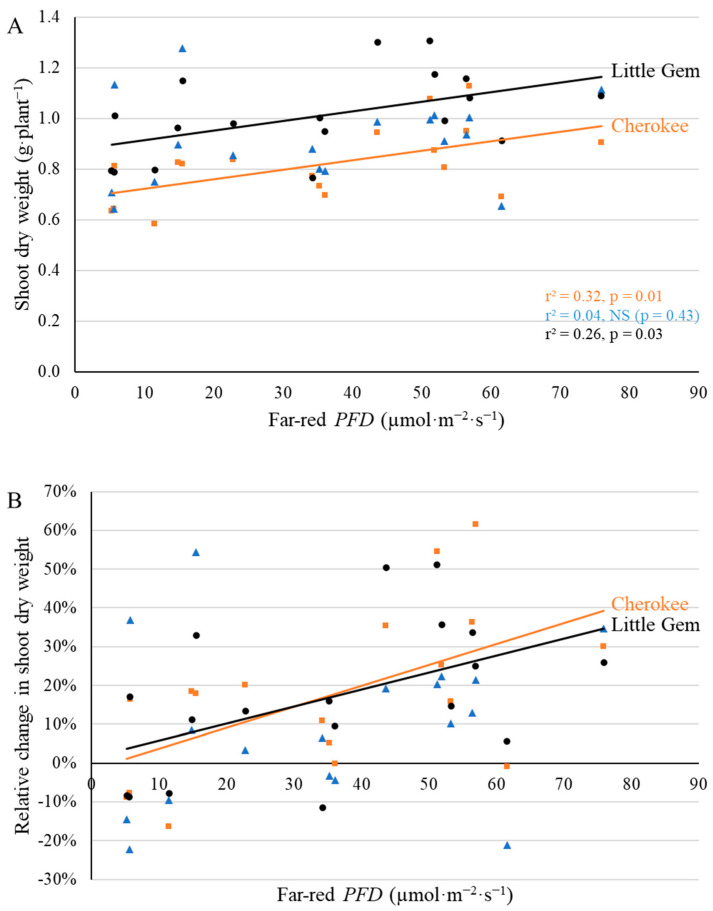
With a background of 204 µmol·m^−2^·s^−1^ white LED light, supplemental far-red photons linearly increased shoot dry weight (**A**) and relative shoot dry weight ((**B**), compared to plants under no supplemental far-red light) at 20 days after the start of far-red light treatment for ‘Cherokee’ and ‘Little Gem’, but not for ‘Green Saladbowl’ (blue triangles) lettuce. The *r*^2^ and *p* values for regression of each cultivar apply to both graphs and are shown with the respective color.

**Figure 12 plants-11-02714-f012:**
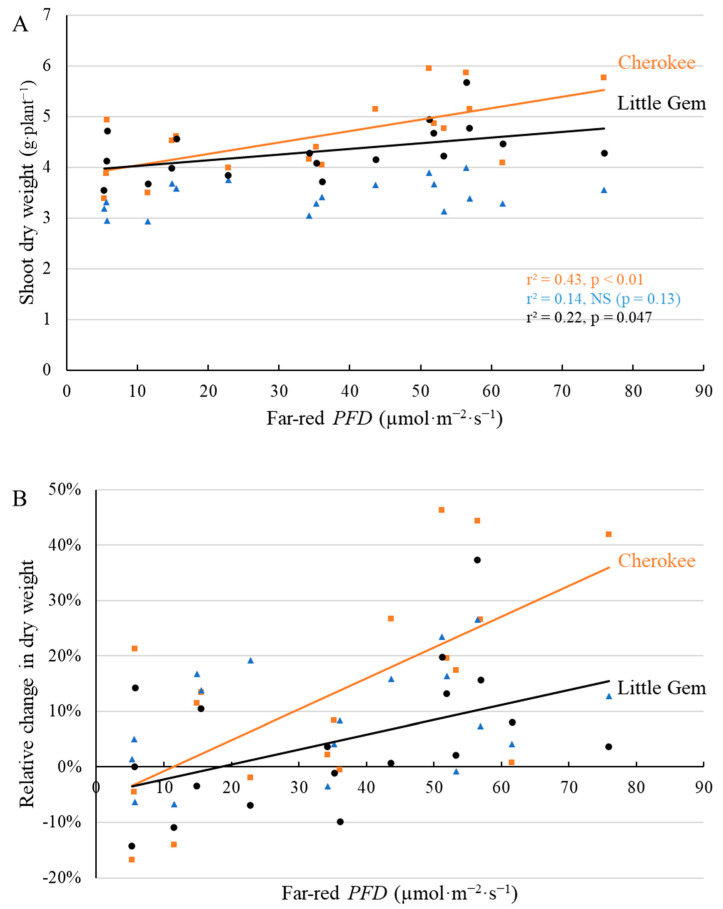
With a background of 204 µmol·m^−2^·s^−1^ white LED light, supplemental far-red photons linearly increased shoot dry weight (**A**) and relative shoot dry weight ((**B**), compared to plants under no supplemental far-red light) at 35 days after the start of far-red light treatment for ‘Cherokee’ and ‘Little Gem’, but not for ‘Green Saladbowl’ (blue triangles) lettuce at 29 DAT. The *r*^2^ and *p* values for regression of each cultivar apply to both graphs and are shown with the respective color.

**Figure 13 plants-11-02714-f013:**
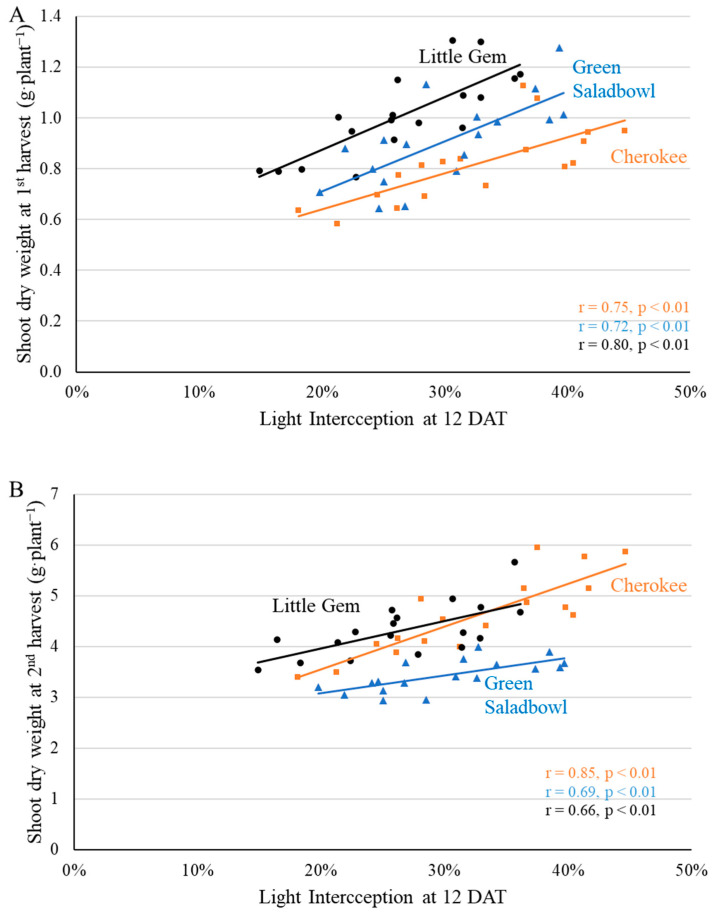
Canopy light interception at an early growth stage (12 DAT) strongly correlated with shoot dry weight at both harvests for ‘Cherokee’, ‘Green Saladbowl’, and ‘Little Gem’ lettuce ((**A**): 1st harvest, (**B**): 2nd harvest). The *r* and *p* values for regression of each cultivar are shown with the respective color.

## Data Availability

The raw data supporting the conclusions of this article will be made available by the authors, without undue reservation.

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
