# Peer review of "Far-Red Light Effects on Lettuce Growth and Morphology in Indoor Production Are Cultivar Specific"

_plants, 2022, doi:10.3390/plants11202714_

Round 1
Reviewer 1 Report
The paper "Far-Red Light Effects on Lettuce Growth and Morphology in Indoor Production are Cultivar-Specific" evaluates the effect of supplying different levels of far-red light to three lettuce cultivars that grow under indoor conditions.
The draft is in general well however the organization of the manuscript must be improved since they do not follow the sequence: introduction, material and methods, results, discussion and conclusions.
In L126-132 and L531-536 when comparing with table S1, some inconsistencies appear, particularly in the description of the control treatments. I suggest unifying the two paragraphs into one and including it in material and methods.
L:126-132 “All plants received warm white LED light (Fig. S1) at 204 ± 11 μmol·m-2·s-1, which included 5.6 ± 0.3 μmol·m-2·s-1 far-red photons. Supplemental far-red light (peak at 724 nm, with a full width at half maximum of 18 nm) was provided ranging from 0 to 70.4 μmol·m-2·s-1, starting 6 days after seeding. Fifteen plants of each cultivar were grown in each of the 18 growing areas. We harvested seven plants at the first harvest (20 DAT) when canopies were closed, and another seven plants at the second harvest at 29 DAT (‘Green Saladbowl’) and 35 DAT (‘Cherokee’ and ‘Little Gem’) after the canopy had closed again.”
L:531-536 Seeds were germinated under white LED light (Fig. S1) of 204 ± 11.0 μmol·m-2·s-1 with a 16 h photoperiod. After germination (6 days after sowing), far-red LED lights were turned on. Plants in three of the 18 units were grown under white LED light only as a control, while plants in the remaining 15 units were grown with supplemental far-red PFDs ranging from 6.0 to 70.3 μmol·m-2·s-1 (Table S1). Far-red light had the same photoperiod as the white LED light.
Minor comments are highlighted in the draft.

Author Response
We thank the reviewer for the careful review.
The required format of journal Plants however follows: introduction, results, discussion, and material and methods, with optional conclusions section, if it is unusually long. We therefore followed the required format.
Since the results section comes before the materials and methods, we decided to provide some details of experiment procedure at the start of the results section to help readers better understand the data we presented here. We thank the reviewer for catching the inconsistencies and edited the text to be more consistent.
The comments made directly in the PDF were all addressed.
Reviewer 2 Report
The paper deals with te effect of supplemental far red light on a number of morphometric and physiological parameters in different cultivars of lettuce.
The papwer, in my opinion, is fine.
However, the experimental approach used by authors did not test any particular working hypothesis, but points to collect a number of (not always related) parameters aimed to describe what is going on under particular growth conditions. In view of the aim of the paper, this is correct but, because of this approach, the paper,in my opinion, could be better collocated in a agronomic journal.
Author Response
The reviewer had no suggested edits, but commented that the paper perhaps should be in an agronomic journal. However, we submitted this article to a special issue on The Effects of LED Light Spectra and Intensities on Plant Growth 2.0 which seems a perfect match for this manuscript.
The reviewer also mentioned 'collect a number of (not always related) parameters aimed to describe what is going on under particular growth conditions' and 'the experimental approach used by authors did not test any particular working hypothesis'. We disagree and feel that our measurements are directly relevant to the stated hypothesis that far-red light can increase growth through morphological (leaf size, canopy light interception) and physiological (photosynthesis) effects.